# No evidence that partnered and unpartnered gay men differ in their preferences for male facial masculinity

**Rachel Cassar** (ID)*, **Victor Shiramizu** (ID), **Lisa M. DeBruine, Benedict C. Jones**

Institute of Neuroscience & Psychology, University of Glasgow, Glasgow, Scotland, United Kingdom

* Rachel.Cassar@glasgow.ac.uk

**Data Availability Statement:** Data, analysis code, and full results output are publicly available at https://osf.io/c4b2r/.

## Abstract

Women's preferences for masculine characteristics in men's faces have been extensively studied. By contrast, little is known about how gay men respond to masculine facial characteristics. One area of disagreement in the emerging literature on this topic is the association between gay men's partnership status and masculinity preference. One study found that partnered gay men showed stronger preferences for masculine faces than did single gay men, while another study found that partnered gay men showed weaker preferences for masculine faces than did single gay men. We re-examined this issue in a sample of 618 gay men, finding no significant difference between partnered and single gay men's masculinity preferences. Together with the mixed previous findings, our null result suggests that the effect of partnership status on gay men's face preferences is not robust.

## Introduction

Exaggerating masculine characteristics in images of men's faces increases perceptions of their dominance and aggressiveness, while decreasing perceptions of their trustworthiness and emotional warmth [1,2]. Given both groups of personality traits can be valuable in a romantic partner [2], many studies have investigated factors that might influence how heterosexual women resolve this apparent trade-off between the potential advantages and disadvantages of choosing a masculine partner. For example, heterosexual women who are currently in a romantic relationship tend to show stronger preferences for masculinized versions of male faces than do heterosexual women not currently in a romantic relationship [3,4, but see also 5]. Such effects are thought to occur because partnered women are less motivated to secure mates with prosocial traits, but may still seek masculine short-term mates who can father healthier and/or more dominant offspring [4].

Most research on preferences for masculinity in men's faces has focused on heterosexual women's preferences. However, a smaller literature has emerged recently examining gay men's preferences for masculinity in men's faces. For example, studies have reported that gay men reporting lower sexual desire [6] or who prefer the penetrative role during intercourse [7,8] show weaker preferences for masculinity in men's faces.

One intriguing inconsistency in this emerging literature on gay men's preferences for masculinity in men's faces concerns the association between partnership status (being in a

**Funding:** European Research Council award to LMD (KINSHIP). The funders had no role in study design, data collection and analysis, decision to publish, or preparation of the manuscript.

**Competing interests:** The authors have declared that no competing interests exist.

relationship versus not being in a relationship) on masculinity preferences. While Zheng [9] found that partnered Chinese gay men showed stronger preferences for masculinity in men's faces than did unpartnered Chinese gay men (similar to findings regarding partnered heterosexual women), Valentova et al. [10] found that partnered Czech gay men showed weaker preferences for masculinity in men's faces than did unpartnered Czech gay men.

In light of the above, the current study compared the masculinity preferences of 432 partnered gay men and 186 unpartnered gay men. Preferences for masculinity in men's faces were assessed using the same procedure as previous research on gay men's face preferences.

## Methods

### Participants

Participants for the online study, which was run at faceresearch.org, were 432 unpartnered men and 186 partnered men aged between 18 and 49 (mean age = 26.42 years, SD = 7.41 years). All men reported that their preferred sex of partner was male. No other exclusion or inclusion criteria were applied. Of the 400 men who reported their country of residence, 2 resided in Africa, 14 resided in Asia, 116 resided in Europe, 232 resided in North America, 20 resided in South America, and 15 resided in New Zealand or Australia. All participants provided informed consent and all procedures were approved by the Psychology Ethics Committee (University of Glasgow).

### Stimuli

Following previous studies of individual differences in women's preferences for masculine faces [3,11], we used prototype-based image transformations to objectively manipulate sexual dimorphism of 2D shape in face images. First, male and female prototype (i.e. average) faces were manufactured using established computer graphic methods that have been widely used in studies of face perception [12]. These prototypes were manufactured using face images of 20 young White male adults and 20 young White female adults, respectively. Next, 50% of the linear differences in 2D shape between symmetrized versions of the male and female prototypes were added to or subtracted from face images of 20 young White male adults. This process created masculinized and feminized versions of the individual face images that differ in sexual dimorphism of 2D shape and that are matched in other regards. Stimuli are publicly available [13]. Example stimuli are shown in Fig 1.

### Procedure

Participants were shown the 20 pairs of face images and were asked to choose the face in each pair that was more attractive. Participants also indicated the strength of these preferences by choosing from the options 'slightly more attractive', 'somewhat more attractive, 'more attractive', and 'much more attractive'. The order in which pairs of faces were shown was fully randomized and the side of the screen on which any particular image was shown was also fully randomized. Responses were coded using a 0 (masculinized face judged as much more attractive than feminized face) to 7 (feminized face judged as much more attractive than masculinized face). These preference scores were centered on chance before being used in our analyses. Each participant also reported their partnership status by answering the question "Do you have a partner? (e.g. a boyfriend, girlfriend, husband, wife, etc.)" and reported the sex of their current partner.

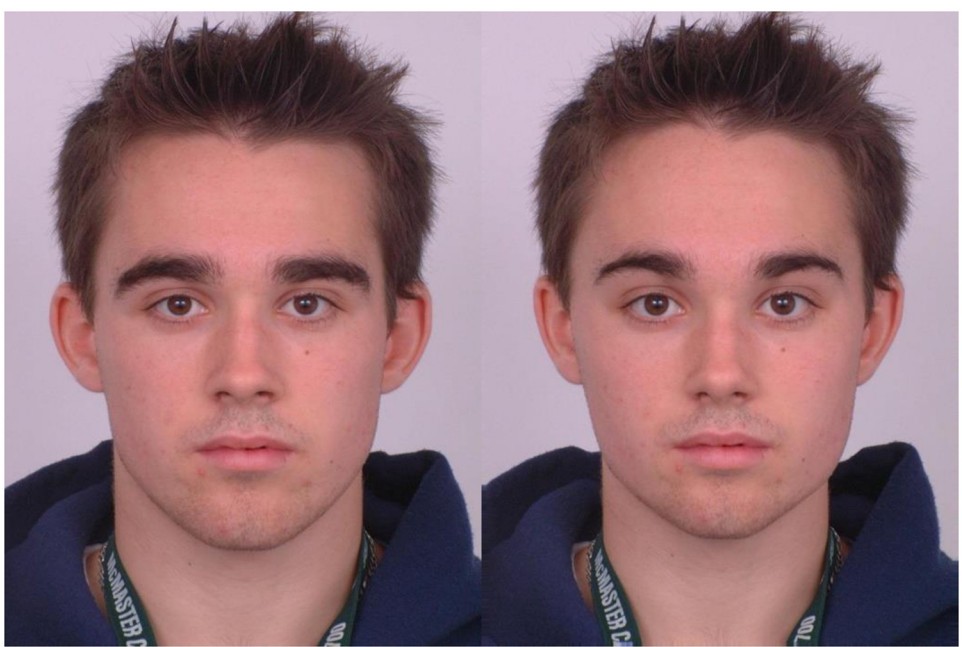

**Fig 1. Examples of masculinized and feminized male faces used in the study.** Stimuli are publicly available [13].

## Results

Analyses were conducted using R v3.4.3. Data, analysis code, and full results output are publicly available at https://osf.io/c4b2r/. First, we analyzed preference scores using a mixed effect model using lmer and lmerTest [14,15] with the between-subject factor *partnership status* (effect coded partnered = 0.5, unpartnered = -0.5), and the covariate *participant age* (centered and scaled on mean of sample). Random intercepts were included for participant and stimulus, with random slopes specified maximally [16,17]. In line with previous research on effects of partnership status on masculinity preferences, we did not include the interaction between partnership status and participant age in our model.

The intercept was significant and negative, indicating that the men in our study generally preferred masculinized faces to feminized faces (estimate = - 0.398, SE = 0.122, df = 23.172, t = - 3.270, p = 0.003). The effect of partnership status was not significant (estimate = 0.114, SE = 0.081, df = 608.211, t = 1.417, p = 0.157). Although older men tended to have weaker preferences for feminized male faces than did younger men, the effect of participant age was not significant (estimate = -0.072, SE = 0.039, df = 280.411, t = -1.842, p = 0.066). An identical model, but this time without any random slopes, showed a qualitatively similar pattern of results (see https://osf.io/c4b2r/).

## Discussion

We found that gay men generally (i.e. on average) preferred masculinized versions of male faces to feminized versions. This preference for masculinity is consistent with Glassenberg et al. [18], who also found that gay men generally preferred masculinized versions of male faces.

We found no effect of partnership status on gay men's masculinity preference. Thus, we did not replicate the negative effect of partnership status on facial masculinity preferences reported for gay Czech men by Valentova et al. [10] or the positive effect of partnership status on facial masculinity preferences reported for gay Chinese men by Zheng [9]. Together, these mixed

results suggest that partnership status does not have a robust effect on gay men's masculinity preferences.

Some work suggests that findings for forced choice preferences (the type of preferences we assessed in the current study) do not necessarily generalize to studies using rating paradigms [19]. Since Valentová et al. [10] used a rating paradigm, this type of paradigm-contingent difference might explain why we did not replicate the effect of partnership status that they reported. However, since Zheng [9] also used a forced choice paradigm, this issue cannot explain why we did not replicate Zheng's results. It is possible that the effect of partnership on gay men's face preferences previously reported are not robust, potentially because same-sex couples are not affected by the putative heritable benefits of choosing a masculine partner. Alternatively, the differences in results across these studies could mean that effects of partnership status on masculinity preferences are somewhat culture-specific.

Previous research on straight and gay men's mate preferences has suggested that both groups show similarities in their mate preferences [20]. For example, both straight and gay men prioritize good looks over other traits when choosing partners [20]. Little et al. [21] reported that partnered straight men showed stronger preferences for feminine women than did unpartnered women. Together with the null result for partnership status in the current study, these findings suggest that partnership status may have different effects on straight and gay men's mate preferences.

Finally, we found that older gay men tended to have stronger masculinity preferences. Although not significant in the current study (p = .067), this relationship is consistent with a general pattern of results whereby older individuals show stronger preferences for facial masculinity, potentially because more masculine faces appear older [1].

We found no evidence that partnership status moderates gay men's preferences for masculine faces. Together with previous research reporting either positive or negative effects of partnership status on gay men's masculinity preferences, these results suggest that partnership status does not have a reliable or robust effect on gay men's preferences for masculine faces.

## Author Contributions

**Conceptualization:** Rachel Cassar, Victor Shiramizu, Lisa M. DeBruine, Benedict C. Jones.

**Formal analysis:** Rachel Cassar, Lisa M. DeBruine.

**Methodology:** Rachel Cassar, Lisa M. DeBruine, Benedict C. Jones.

**Resources:** Lisa M. DeBruine, Benedict C. Jones.

**Software:** Lisa M. DeBruine.

**Supervision:** Lisa M. DeBruine, Benedict C. Jones.

**Writing – original draft:** Rachel Cassar, Victor Shiramizu, Lisa M. DeBruine, Benedict C. Jones.

**Writing – review & editing:** Rachel Cassar, Victor Shiramizu, Lisa M. DeBruine, Benedict C. Jones.

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
