## [Decision Letter · Decision Letter 0]

13 Jan 2020

PONE-D-19-22159

No evidence that partnered and unpartnered gay men differ in their preferences for male facial masculinity

PLOS ONE

Dear Ms Cassar,

Thank you for submitting your manuscript to PLOS ONE. After careful consideration, we feel that it has merit but does not fully meet PLOS ONE’s publication criteria as it currently stands. Therefore, we invite you to submit a revised version of the manuscript that addresses the points raised during the review process.

Please accept my apologies in returning your manuscript to you. The original handling editor was unable to continue with the submission, and it has been passed to me. I had to take a few days to acquaint myself with the manuscript and the comments provided by two reviewers.

As you will see, these comments are generally favourable - both reviewers found much to like in the manuscript, and there is great value in the use of large sample sizes to further our understanding of these effects. While there are specific points from each reviewer that I felt should be attended to (e.g., the origin of the raters and whether this is worth modelling), the overarching theme of the reviews was that the manuscript was presented in a 'theory-light' manner. I am inclined to agree with the reviewers on this point. The introduction is brief and does not have theoretical rationales for why partnered women might prefer masculine faces, why current findings have conflicting results and why this might be, and how the null effect here might sit within the wider field. Both reviewers made some important points regarding this that would significantly strengthen the manuscript if addressed. In particular, the differences in experimental design of studies examining similar questions could be discussed in detail. Manipulations and forced-choice presentation of faces to be selected for attractiveness can show effects that disappear under more naturalistic viewing conditions (Jones & Jaeger, 2019; Symmetry), and this could be worth discussing here.

Finally, I had my own comments regarding the model specification in the analysis. It was stated that maximal slopes were included, and I can see that the variance-covariance structure contained slopes for participant + age with face ID, but this was given without much justification aside from a citation. However, I thought it was unusual to not include an interaction term here between age and partner status, which would certainly fall under the 'maximal' rule of specifying a model that captures the experimental data-generating process fully - and one can imagine there may be evolutionary relevant outcomes for preferring masculinity as a function of age and partnership status.

However, though the 'keep it maximal' approach is very popular within psychology, a different approach recommended by others is to build simpler, more parsimonious models that are not over-specified (https://arxiv.org/abs/1506.04967). It is not straightforward to quantitatively label over specification in mixed models, but running the analysis for myself showed me the model failed to converge, and the variance of partner and age under face ID was essentially zero, which is a symptom of over specification. I would encourage the authors to reconsider their specification here and build up from a simpler model (looking at this data, it doesn't seem that there is benefit beyond random intercepts for both faces and participants).

Taken together, I would like to invite the authors to submit a revised version of the manuscript, and look forward to receiving it.

Very best wishes

We would appreciate receiving your revised manuscript by Feb 24 2020 11:59PM. To enhance the reproducibility of your results, we recommend that if applicable you deposit your laboratory protocols in protocols.io, where a protocol can be assigned its own identifier (DOI) such that it can be cited independently in the future. For instructions see: http://journals.plos.org/plosone/s/submission-guidelines#loc-laboratory-protocols

We look forward to receiving your revised manuscript.

Kind regards,

Alex Jones

Academic Editor

PLOS ONE

Journal Requirements:

2. In order to meet our 3rd publication criterion (https://journals.plos.org/plosone/s/criteria-for-publication#loc-3) we would be grateful if you could please provide further details about the your experimental procedures.

Specifically, we would be grateful if you could specify:

A) Details of any inclusion and exclusion criteria applied to participants in the study.

B) Details of how relationship status was determined.

Please provide details of how participants were coded as being not/ in a relationship. For example, could participants respond as being in an open relationship? If so, how was this coded in your study?

3. Please also provide a source for the image in Figure 1 in the figure legend.

Reviewers' comments:

Reviewer's Responses to Questions

**Comments to the Author**

1. Is the manuscript technically sound, and do the data support the conclusions?

Reviewer #1: Partly

Reviewer #2: Yes

2. Has the statistical analysis been performed appropriately and rigorously? 

Reviewer #1: Yes

Reviewer #2: Yes

3. Have the authors made all data underlying the findings in their manuscript fully available?

Reviewer #1: Yes

Reviewer #2: Yes

4. Is the manuscript presented in an intelligible fashion and written in standard English?

Reviewer #1: Yes

Reviewer #2: Yes

5. Review Comments to the Author

Reviewer #1: In general, I liked the manuscript, the idea is good and given the ambiguous results from the only two previous studies on the same topic, a new well-done study is more than welcome. I have a few suggestions to improve the manuscript:

1. In general, I miss theoretical base for the current research. For example, Introduction, 1st paragraph – “heterosexual women who are currently in a romantic relationship tend to show stronger preferences for masculinized versions of male faces than do heterosexual women not currently in a romantic relationship”. What is the theoretical reasoning behind these findings? Why should partnered women prefer rather masculine male faces?

2. Introduction, 2nd paragraph – Why is homosexual sample so interesting in the research on partner preferences? Some theoretical note?

3. Introduction, 2nd paragraph – why the two studies showed different results? They employed different methods, and they were done in different populations. This might be a very important discussion, and can appear either in the Introduction or in the Discussion of the current manuscript.

4. Methods, Participants – do the authors know where were the raters from? From the manuscript it seems they indicated country of residence, so the sample is a mixture of people from several continents. This might be the answer why no specific tendency of masculine versus feminine facial preferences appeared in the present study. If these preferences differ among populations, then the population should be included into the analysis.

5. Methods, Stimuli – it should be noted that that the study design was similar to Zheng et al (2019), but different from Valentova et al (2013).

6. Results – I guess that there is no need for five decimal places, two or three would be enough.

7. Discussion – it seems to me that this section actually does not discuss the current findings. It only affirms that the results are different from the two previous studies on a similar topic, but it does not try to explain why it is so. Again, it may be methodological differences (at least difference from Valentova et al, 2013 who used natural photos, while in the current study and Zheng 2019 manipulated facial pictures were employed), but the different results can be also caused by different populations. This is the minimum required for a discussion, although I was expecting more. The literature on preferences of homosexual individuals is rare, and any new material should bring not only methodological advance but also some theoretical reasoning.

Reviewer #2: The article is well-presented and very easy to understand - obviously an N of 618 is a big advantage too. I feel like it would have benefited from more discussion though - namely, how this paper adds to the literature on how different genders perceive and value physical attractiveness. For example, in 'The Evolution of Desire', Buss (2003: 60-3) presents studies which argue that homosexual men pattern with heterosexual men in the value they place on physical attractiveness, whereas homosexual women pattern with heterosexual women (i.e. straight and gay men value 'good looks' more than women of any orientation). This paper seems to make a similar argument - that partnership status influences what women find physically attractive, but this is not the case for (homosexual) men. Are there any papers that investigate whether heterosexual men prefer feminine faces when they're single/in a relationship? If it's established that straight men show no different preferences for femininity based on their relationship status, that would make a nice companion to this study. I also wonder whether homosexual women are more attracted to feminine faces when they're in a relationship?

In other words, does partnership status have a robust effect of the preferences of women (of any orientation) but not men? And could this article be integrated into the wider theory regarding this?

Additionally, I would cut the reported statistics down to 3 significant figures to make some sections more readable and include the random effects estimates in the text.

6. PLOS authors have the option to publish the peer review history of their article (what does this mean?). If published, this will include your full peer review and any attached files.

Reviewer #1: Yes: Jaroslava Varella Valentova

Reviewer #2: No

---

## [Author Response · Author response to Decision Letter 0]

30 Jan 2020

Editor’s comments

As you will see, these comments are generally favourable - both reviewers found much to like in the manuscript, and there is great value in the use of large sample sizes to further our understanding of these effects. While there are specific points from each reviewer that I felt should be attended to (e.g., the origin of the raters and whether this is worth modelling), the overarching theme of the reviews was that the manuscript was presented in a 'theory-light' manner. I am inclined to agree with the reviewers on this point. The introduction is brief and does not have theoretical rationales for why partnered women might prefer masculine faces, why current findings have conflicting results and why this might be, and how the null effect here might sit within the wider field. Both reviewers made some important points regarding this that would significantly strengthen the manuscript if addressed. In particular, the differences in experimental design of studies examining similar questions could be discussed in detail. Manipulations and forced-choice presentation of faces to be selected for attractiveness can show effects that disappear under more naturalistic viewing conditions (Jones & Jaeger, 2019; Symmetry), and this could be worth discussing here.

We now discuss this issue in our Discussion: “Some work suggests that findings for forced choice preferences (the type of preferences we assessed in the current study) do not necessarily generalize to studies using rating paradigms [19]. Since Valentová et al. [10] used a rating paradigm, this type of paradigm-contingent difference might explain why we did not replicate the effect of partnership status that they reported. However, since Zheng [9] also used a forced choice paradigm, this issue cannot explain why we did not replicate Zheng’s results. It is possible that the effect of partnership on gay men’s face preferences previously reported are not robust, potentially because same-sex couples are not affected by the putative heritable benefits of choosing a masculine partner. Alternatively, the differences in results across these studies could mean that effects of partnership status on masculinity preferences are somewhat culture-specific.”

Finally, I had my own comments regarding the model specification in the analysis. It was stated that maximal slopes were included, and I can see that the variance-covariance structure contained slopes for participant + age with face ID, but this was given without much justification aside from a citation. However, I thought it was unusual to not include an interaction term here between age and partner status, which would certainly fall under the 'maximal' rule of specifying a model that captures the experimental data-generating process fully - and one can imagine there may be evolutionary relevant outcomes for preferring masculinity as a function of age and partnership status.

However, though the 'keep it maximal' approach is very popular within psychology, a different approach recommended by others is to build simpler, more parsimonious models that are not over-specified (https://arxiv.org/abs/1506.04967). It is not straightforward to quantitatively label over specification in mixed models, but running the analysis for myself showed me the model failed to converge, and the variance of partner and age under face ID was essentially zero, which is a symptom of over specification. I would encourage the authors to reconsider their specification here and build up from a simpler model (looking at this data, it doesn't seem that there is benefit beyond random intercepts for both faces and participants).

After extensive consideration and discussion, we have opted to retain our (a priori) analysis plan, but have clarified why the interaction between partnership status and age is not included in our model (Results: “In line with previous research on effects of partnership status on masculinity preferences, we did not include the interaction between partnership status and participant age in our model.”). We note here that the data and code are open, so any readers who want to carry out exploratory analyses with other model specifications can obviously do so.

We have also rerun our main analysis with no random slopes and have added this sentence to our Results: “An identical model, but this time without any random slopes, showed a qualitatively similar pattern of results (see https://osf.io/c4b2r/).”

Details of any inclusion and exclusion criteria applied to participants in the study.

Methods: “Participants for the online study, which was run at faceresearch.org, were 432 unpartnered men and 186 partnered men aged between 18 and 49 (mean age = 26.42 years, SD = 7.41 years). All men reported that their preferred sex of partner was male. No other exclusion or inclusion criteria were applied.

Details of how relationship status was determined.”

Text added to Methods: “Each participant also reported their partnership status by answering the question “Do you have a partner? (e.g. a boyfriend, girlfriend, husband, wife, etc.)” and reported the sex of their current partner.”

Please provide details of how participants were coded as being not/ in a relationship. For example, could participants respond as being in an open relationship? If so, how was this coded in your study?

This was not considered on our study (or in previous research on the topic).

Reviewer #1’s comments

In general, I liked the manuscript, the idea is good and given the ambiguous results from the only two previous studies on the same topic, a new well-done study is more than welcome. I have a few suggestions to improve the manuscript:

1. In general, I miss theoretical base for the current research. For example, Introduction, 1st paragraph – “heterosexual women who are currently in a romantic relationship tend to show stronger preferences for masculinized versions of male faces than do heterosexual women not currently in a romantic relationship”. What is the theoretical reasoning behind these findings? Why should partnered women prefer rather masculine male faces?

We have clarified this point in our Introduction: “Such effects are thought to occur because partnered women are less motivated to secure mates with prosocial traits, but may still seek masculine short-term mates who can father healthier and/or more dominant offspring [4].”

2. Introduction, 2nd paragraph – Why is homosexual sample so interesting in the research on partner preferences? Some theoretical note?

We have clarified this point in our Introduction: “Such findings may offer insight into the extent to which potential heritable benefits (e.g., increased offspring viability) drive mate preferences.”

3. Introduction, 2nd paragraph – why the two studies showed different results? They employed different methods, and they were done in different populations. This might be a very important discussion, and can appear either in the Introduction or in the Discussion of the current manuscript.

We now discuss this issue in our Discussion: “Some work suggests that findings for forced choice preferences (the type of preferences we assessed in the current study) do not necessarily generalize to studies using rating paradigms [19]. Since Valentová et al. [10] used a rating paradigm, this type of paradigm-contingent difference might explain why we did not replicate the effect of partnership status that they reported. However, since Zheng [9] also used a forced choice paradigm, this issue cannot explain why we did not replicate Zheng’s results. It is possible that the effect of partnership on gay men’s face preferences previously reported are not robust, potentially because same-sex couples are not affected by the putative heritable benefits of choosing a masculine partner. Alternatively, the differences in results across these studies could mean that effects of partnership status on masculinity preferences are somewhat culture-specific.”

4. Methods, Participants – do the authors know where were the raters from? From the manuscript it seems they indicated country of residence, so the sample is a mixture of people from several continents. This might be the answer why no specific tendency of masculine versus feminine facial preferences appeared in the present study. If these preferences differ among populations, then the population should be included into the analysis.

We note here that majority of our participants are from western countries (~87%). Consequently, we have not included country as a factor in our model.

5. Methods, Stimuli – it should be noted that that the study design was similar to Zheng et al (2019), but different from Valentova et al (2013).

We now discuss the methodological differences in our Discussion.

6. Results – I guess that there is no need for five decimal places, two or three would be enough.

We have rounded the figures to three places.

7. Discussion – it seems to me that this section actually does not discuss the current findings. It only affirms that the results are different from the two previous studies on a similar topic, but it does not try to explain why it is so. Again, it may be methodological differences (at least difference from Valentova et al, 2013 who used natural photos, while in the current study and Zheng 2019 manipulated facial pictures were employed), but the different results can be also caused by different populations. This is the minimum required for a discussion, although I was expecting more. The literature on preferences of homosexual individuals is rare, and any new material should bring not only methodological advance but also some theoretical reasoning.

We have expanded on these points in our Discussion: “Some work suggests that findings for forced choice preferences (the type of preferences we assessed in the current study) do not necessarily generalize to studies using rating paradigms [19]. Since Valentová et al. [10] used a rating paradigm, this type of paradigm-contingent difference might explain why we did not replicate the effect of partnership status that they reported. However, since Zheng [9] also used a forced choice paradigm, this issue cannot explain why we did not replicate Zheng’s results. It is possible that the effect of partnership on gay men’s face preferences previously reported are not robust, potentially because same-sex couples are not affected by the putative heritable benefits of choosing a masculine partner. Alternatively, the differences in results across these studies could mean that effects of partnership status on masculinity preferences are somewhat culture-specific.”

And: “Previous research on straight and gay men’s mate preferences has suggested that both groups show similarities in their mate preferences [20]. For example, both straight and gay men prioritize good looks over other traits when choosing partners [20]. Little et al. [21] reported that partnered straight men showed stronger preferences for feminine women than did unpartnered women. Together with the null result for partnership status in the current study, these findings suggest that partnership status may have different effects on straight and gay men’s mate preferences.”

Reviewer #2’s comments

The article is well-presented and very easy to understand - obviously an N of 618 is a big advantage too. I feel like it would have benefited from more discussion though - namely, how this paper adds to the literature on how different genders perceive and value physical attractiveness. For example, in 'The Evolution of Desire', Buss (2003: 60-3) presents studies which argue that homosexual men pattern with heterosexual men in the value they place on physical attractiveness, whereas homosexual women pattern with heterosexual women (i.e. straight and gay men value 'good looks' more than women of any orientation). This paper seems to make a similar argument - that partnership status influences what women find physically attractive, but this is not the case for (homosexual) men. Are there any papers that investigate whether heterosexual men prefer feminine faces when they're single/in a relationship? If it's established that straight men show no different preferences for femininity based on their relationship status, that would make a nice companion to this study. I also wonder whether homosexual women are more attracted to feminine faces when they're in a relationship? In other words, does partnership status have a robust effect of the preferences of women (of any orientation) but not men? And could this article be integrated into the wider theory regarding this?

We now discuss this issue in our Discussion: “Previous research on straight and gay men’s mate preferences has suggested that both groups show similarities in their mate preferences [20]. For example, both straight and gay men prioritize good looks over other traits when choosing partners [20]. Little et al. [21] reported that partnered straight men showed stronger preferences for feminine women than did unpartnered women. Together with the null result for partnership status in the current study, these findings suggest that partnership status may have different effects on straight and gay men’s mate preferences.”

Additionally, I would cut the reported statistics down to 3 significant figures to make some sections more readable and include the random effects estimates in the text.

We have rounded the figures to three places.

---

## [Editor Report · Decision Letter 1]

31 Jan 2020

No evidence that partnered and unpartnered gay men differ in their preferences for male facial masculinity

PONE-D-19-22159R1

Dear Dr. Cassar,

We are pleased to inform you that your manuscript has been judged scientifically suitable for publication and will be formally accepted for publication once it complies with all outstanding technical requirements.

With kind regards,

Alex Jones

Academic Editor

PLOS ONE

---

## [Editor Report · Acceptance letter]

21 Feb 2020

PONE-D-19-22159R1 

No evidence that partnered and unpartnered gay men differ in their preferences for male facial masculinity 

Dear Dr. Cassar:

I am pleased to inform you that your manuscript has been deemed suitable for publication in PLOS ONE. Congratulations! Your manuscript is now with our production department. 

With kind regards,

on behalf of

Dr. Alex Jones 

Academic Editor

PLOS ONE